# Youth Culturally adapted Manual Assisted Problem Solving Training (YCMAP) in Pakistani adolescent with a history of self-harm: protocol for multicentre clinical and cost-effectiveness randomised controlled trial

Nusrat Husain [1,2] Sehrish Tofique,[3] Imran B Chaudhry,[4,5] Tayyeba Kiran [6] Peter Taylor,[7] Christopher Williams [8] Rakhshi Memon,[9] Shilpa Aggarwal,[10] Mohsin Hassan Alvi,[11] S Ananiadou,[12] Moin Ahmad Ansari,[13] Saadia Aseem,[14] Andrew Beck,[15] Sumira Alam,[16] Erminia Colucci [17] Kate Davidson,[18] Sarah Edwards,[19] Richard Emsley,[20] Jonathan Green,[21] Anil Gumber,[22] Keith Hawton [23] Farhat Jafri,[24] Ayesha Khaliq,[25] Thomas Mason [26] Ann Mcreath,[27] Ayesha Minhas,[28] Farooq Naeem,[29] Haider Ali Naqvi,[30] Amna Noureen,[31] Maria Panagioti,[32] Anita Patel,[33] Aaron Poppleton,[34] Tinevimbo Shiri,[35] Mima Simic,[36] Sarwat Sultan,[37] Asad Tamizuddin Nizami,[38] Zainab Zadeh,[39] Shehla Naeem Zafar,[40] Nasim Chaudhry[41]

For numbered affiliations see end of article.

**Correspondence to**
Tayyeba Kiran;
tayyebakiran@gmail.com

## ABSTRACT

**Introduction** Suicide is a global health concern. Sociocultural factors have an impact on self-harm and suicide rates. In Pakistan, both self-harm and suicide are considered as criminal offence's and are condemned on both religious and social grounds. The proposed intervention 'Youth Culturally Adapted Manual Assisted Problem Solving Training (YCMAP)' is based on principles of problem-solving and cognitive–behavioural therapy. YCMAP is a brief, culturally relevant, scalable intervention that can be implemented in routine clinical practice if found to be effective.

**Method and analysis** A multicentre rater blind randomised controlled trial to evaluate the clinical and cost-effectiveness of YCMAP including a sample of 652 participants, aged 12–18 years, presenting to general physicians/clinicians, emergency room after self harm or self referrals. We will test the effectiveness of 8–10 individual sessions of YCMAP delivered over 3 months compared with treatment as usual. Primary outcome measure is repetition of self-harm at 12 months. The seconday outcomes include reduction in suicidal ideation, hopelessness and distress and improvement in health related quality of life. Assessments will be completed at baseline, 3, 6, 9 and 12 months postrandomisation. The nested qualitative component will explore perceptions about management of self-harm and suicide prevention among adolescents and investigate participants' experiences with YCMAP. The study will be guided by the theory of change approach to ensure that the whole trial is centred around needs of the end beneficiaries as key stakeholders in the process.

## STRENGTHS AND LIMITATIONS OF THIS STUDY

⇒ This is the first multicentre randomised controlled trial to evaluate clinical and cost-effectiveness of a culturally relevant psychological intervention for young people presenting with self-harm in a low-income and middle-income country.

⇒ Process evaluation will include in-depth interviews to understand the lived experiences of participants.

⇒ Stigma, fear of persecution by authorities, peer pressure and educational commitments may act as a barriers to young people participating in the study.

⇒ The theory of change approach will help mitigate some of the risks around stigma and refusal to participate by involving parents and young people from the inception of the trial.

**Ethics and dissemination** Ethics approval has been obtained from the Ethics Committee of University of Manchester, the National Bioethics Committee in Pakistan. The findings of this study will be disseminated through community workshops, social media, conference presentations and peer-reviewed journals.

**Trial registration number** NCT04131179.

## INTRODUCTION

Suicide is one of the major public health concerns globally with 700 000 suicides each year across the globe.[1] More than 77% of suicides occur in low-income and

middle-income countries (LMICs). However, there are now major concerns about increasing rates of self-harm and suicide among young people in high-income countries such as USA[2] and Europe.[3] Overall, suicide is the second-leading cause of death in individuals between 15 and 29 years old.[1] A recent review reported suicide rates in South Asia to be higher than the global average.[4] However, such figures are likely to be an underestimate due to a lack of accurate data on suicide in Pakistan[5] and in many other LMICs.[6]

In Pakistan, a conservative Islamic state, policies are changing but suicide and self-harm (defined below) remain criminal acts and are condemned both socially and religiously as a moral wrong. Suicide and self-harm are often considered taboo subjects across the country, thus contributing to a lack of evidence and under-reporting.[4] There is, however, accumulating evidence that both self-harm and suicide rates have been increasing in Pakistan but there continues to be notable gaps in evidence.[4] In a systematic review of mental health studies of adolescents in India, it was found that self-harm is particularly problematic in young people, with 3 months prevalence ranging from 3.9% to 25.4% in community-based studies.[7] [8] Self-harm is one of the strongest predictors of death by suicide.[9] Self-harm also carries a substantial economic impact and has been associated with large treatment costs in Pakistan.[10] Adolescents with a history of self-harm are at a higher risk of repeating this later in life,[11] thus there is an important need to develop effective interventions in order to avoid long-term burden.

In adolescents, self-harm is the result of a highly complex interplay between genetic, biological, psychological, social and cultural factors.[11] Common set of risk factors and psycho-social mechanisms for self-harm within LMICs include being female, experiencing interpersonal conflict, suffering from abuse (including domestic/family/gender based violence), hopelessness and being diagnosed with psychiatric disorder.[4] Other psychological influences include feelings of entrapment, a lack of belonging or connection.[12] Research in Pakistan is imperative to explore the effectiveness of interventions for potential risk factors to reduce rates of self-harm and suicide.[13] The WHO Report 'preventing Suicide: A Global imperative' recommends 'twofold' public health approach for suicide prevention, that is, to identify the issues, and provide treatment for high-risk individuals. Considering the benefits of adult individual cognitive–behaviour therapy (CBT)-based psychotherapy,[14] tested in our previous trial in Pakistan with positive results[15] and replicated in our recently completed trial with 901 adults participants in which we had to exclude 265 adolescents (less than 18 years) (Culturally adapted Manual Assisted brief Psychological intervention, CMAP2).[16]

An unmet need was recognised to develop CMAP further for adolescent and adapting the similar brief psychological intervention for children and adolescents. Psychological therapies that work in western culture cannot always be implemented in a different culture and adaptation is considered to be essential[17] [18]

Since interpersonal conflicts with family members are a commonly reported precipitant of self-harm episodes in Pakistan.[4] [13] It has been reported that problem-solving therapy 'CMAP' may be a useful intervention for prevention of self-harm in Pakistan.[13] This psychological intervention, using variants of CBT that are age appropriate for the local culture and customs in Pakistan, may prove to be beneficial in reducing self-harm in adolescents.[15] Before this intervention can be implemented at a larger scale, there is a need for establishing its clinical and cost-effectiveness via the proposed trial. We cannot assume that an intervention developed for adults would be suitable for children and adolescents and since self-harm remains a major problem in young people, this planned trial fills an important gap and clinical need.

## METHODS AND ANALYSIS
### Objectives
This study includes both quantitative and qualitative aspects.

The objectives of the quantitative component are:
1. To determine the clinical effectiveness of Youth Culturally Adapted Manual Assisted Problem Solving Training (YCMAP) over 12 months based on the following outcomes:
   a. Primary outcome:
      i. To measure the effectiveness of YCMAP in comparison to treatment as usual (TAU) in terms of repetition of self-harm 12-month postrandomisation. This will be assessed using adapted Suicide Attempt Self-Injury Interview (SASII).[19]
   b. Secondary outcomes:
      i. Suicidal ideation will be assessed using the 'Beck Scale for Suicidal ideation'.[20]
      ii. Feelings of hopelessness will be assessed using the 'Beck Hopelessness Scale'.[21]
      iii. Level of distress will be assessed using a 10-item scale the 'Kessler Psychological Distress Scale'.[22]
      iv. EQ-5D- Y will be used to assess 'health-related quality of life'.[23]
      v. Client Satisfaction Questionnaire-8 will be used to assess participants' satisfaction with services.[24]

Translated versions of all these scales have been developed and used in previous trials CMAP1 (15) and CMAP 2 (13) by following established protocols for translating such measures.[25] The scales which were not available in Urdu were translated for an MPhil project (Brief psychological intervention for adolescents who self harm, IPP/BU/ER/103/1377) and reviewed in initial Patient and Public Involvement and Engagement (PPIE) meetings.

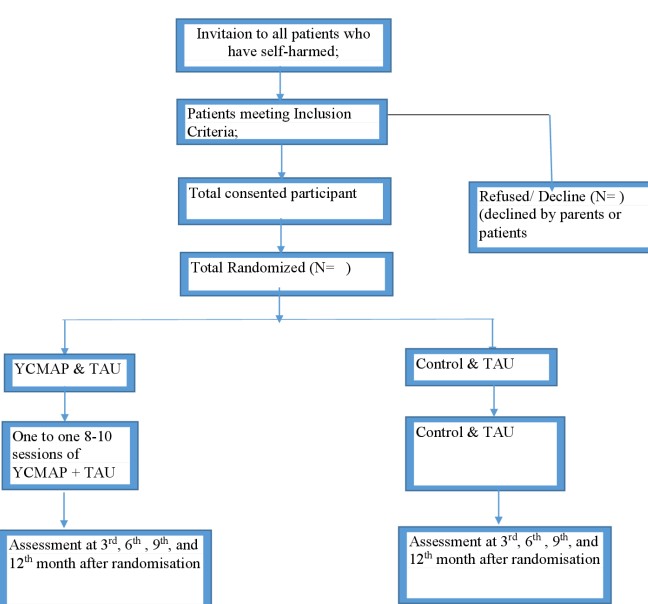

**Figure 1** Flow diagram of the YCMAP study. TAU, treatment as usual; YCMAP, Youth Culturally Adapted Manual Assisted Problem Solving Training.

Assessments will be carried out at baseline, completion of the intervention (3 months), 6, 9 and 12 months after randomisation. See figure 1 below

2. To determine the cost-effectiveness of the YCMAP intervention over 12 months.
   The objectives of the qualitative component are:
1. To explore the experience of participants with YCMAP and their perception about benefits and negative or adverse consequence of it.
2. To explore in detail participants' reasons for continuing (completer) or not continuing (drop out) with the study.
3. To explore clinicians and other stakeholder views (such as school teachers) on the YCMAP and its impact.
4. To explore therapists' perspective on the delivery of the YCMAP.
   The expected outputs of the study are:
1. A manual-assisted evidence based intervention—the YCMAP, ready for integration into the healthcare services.
2. Standardised training and supervision package—a resource pack and training programme on 'how to do it' for use across health services.
3. Information tool-kit for the families and the wider community. This will be service-user defined and will be used to increase awareness about mental health in general and self-harm and suicide in young people in particular.

## Inclusion criteria
Adolescents meeting the study inclusion criteria will be invited to take part in the study by their primary care clinician, ward or emergency room clinician who is making the initial assessment. In the context of this study, self-harm is defined as: 'an act with non-fatal outcome, in

which an individual deliberately initiates a non-habitual behaviour that, without interventions from others, will cause self-harm, or deliberately ingests a substance in excess of the prescribed or generally recognised therapeutic dosage, and which is aimed at realising changes which the subject desired via the actual or expected physical consequences'.[26]

► Age: 12–18 years.
► History of recent self-harm which is defined as 'self-harm occurring within the last 3 months (from the initial identification of a potential participant)'. This period is considered as high risk for repetition in young people.[11]
► Resident in the trial catchment area.
► All participants known to clinical or health services. This will be necessary to ensure all participants have access to TAU.

## Exclusion criteria
► Patients with a severe mental illness such as schizophrenia spectrum and other psychotic disorders, bipolar and severe depressive disorder will be excluded. This intervention has been developed to help individuals who experience common mental health problems such as anxiety or mild to moderate depressive disorders. It has not been developed to work with people with severe mental health difficulties such as psychotic disorders. We consider that the intervention would require work with service users and carers and further adaptation work with people who have severe mental health illness.
► Patients with conditions limiting engagement with assessment/intervention.
► Temporary resident unlikely to be available for follow-up.

## Design
A multicentre rater blind randomised controlled trial (RCT) with randomisation by individual participants in order to compare the YCMAP in addition to TAU with TAU alone.

## Study site and population
Study sites are all participating primary care clinics, emergency departments and medical wards of general hospitals in five major cities of Pakistan, Karachi, Hyderabad, Lahore, Multan and Rawalpindi.

## Sample size
The sample size is based on the primary outcome, repetition of self-harm in a 12-month period (yes/no). The TAU arm of the study has an expected self-harm rate of 20%.[7] A clinically important effect would be a reduction to 7.5% in the intervention group. Under these presumptions, and assuming a 5% significance level and 90% power, a study with no clustering would require 158 patients per arm. The study has a partially nested design due to therapist clustering in the YCMAP arm. Our sample size calculation has taken account of this by adjusting the sample

size upwards, conservatively assuming clustering in both arms. Based on previous analysis of therapist trials, we believe that the intraclass correlation coefficient (ICC) is likely to have a value between 0.01 and 0.05 for this type of outcome measure. Assuming an ICC of 0.05, and a cluster size of 16 patients per therapist, a design effect of 1.75 is calculated. This increases the numbers required to 277 per arm. Furthermore,based on our previous work there is expected to be a 15% loss to follow-up, and so the final numbers recruited will be 326 per arm, with a total sample of 652 participants.

## Randomisation

The researcher will complete a checklist, confirming eligibility, obtain written (or verbal audiorecorded consent from the subject due to current pandemic COVID-19 SOPs), and complete the baseline measures. The researcher will contact the randomisation centre, who will recheck eligibility, record baseline measures and assign a participant trial number. Treatment assignment will then be determined using stochastic minimisation controlling for gender, age and type of self-harm behaviour. The independent statistician would inform the project manager of the random assignments, but this information would not be shared with research staff, so that they remain unaware of whether a particular participant receives YCMAP or not. Participants assigned to the YCMAP would then be contacted by a therapist, who will arrange an initial meeting within 2–4 weeks after baseline to start the intervention. Participants in the TAU group would be informed of their allocation to this group after randomisation has taken place.

## Recruitment and baseline assessment

Potentially eligible individuals will be identified by clinical staff at the participating recruitment sites. An age appropriate version of the participant information sheet (PIS) will be provided alongside the standard PIS, so both the parent/guardian and child can access this information. For individuals who cannot read, information about the study will be summarised by the clinician for them. Consent will be obtained from the parent/guardian along with consent from the participating young person (The Nuffield Council for Bioethics guidance suggests that consent rather than assent should be sought from young people in research). Therefore, it is ensured and clearly mentioned in the PIS for both the young people and their parents that it is mandatory for parents/guardians to consent for young person to participate in the study.

## Intervention

YCMAP is a 'Youth culturally adapted manual assisted psychological intervention' based on CBT principles. It comprises of 8–10 sessions delivered over 3 months. The first eight sessions are offered weekly and further sessions fortnightly on a one to one basis and each session lasts for about 60 min. The YCMAP has been culturally adapted with permission from 'CMAP',[15] 'Life after self-harm'[27] and 'Cutting down: A CBT workbook for treating young people who self-harm'.[28] The intervention includes psycho-education and a comprehensive cognitive behavioural assessment of the self-harm attempt using virtual stories of four young people. The therapy focuses on current problems that contributed to the self-harm episode. Therapists and adolescent clients choose from a list of techniques those which are most relevant to the client's problems. Therapy is therefore adapted to fit with the clients problems and primarily utilises problem solving, CBT and dialectical therapy strategies to bring about change.

To help determine the most appropriate coping strategy a coping tree is designed. Training in assertiveness and anger management are offered to help the young person to develop resilience to cope with stress. To the existing YCMAP we will also aim to review the look and feel of the resource by including licensed content from My Big Life—a course developed by our collaborative partner Five Areas Limited. This accessible, story-based approach uses illustrations of young people facing different scenarios at home and at school to illustrate key CBT-based concepts: Understanding your feelings-How to get a Big Life (behavioural activation)—Thinking in a Big Life way (identifying and changing thoughts that upset and affect how you feel)—Relaxation approaches—Building inner confidence—Practice scenarios—Trainer notes and linked worksheets/prompt cards and posters. Asian versions of the course already exist with amended artwork. As part of the development programme selected content will be added to the course and trainer notes will be modified (based on feedback) as needed.

All research staff will be required to have at least an undergraduate level degree in a relevant area (eg, psychology) and will be trained in 'Good Clinical Practice' in research, consent process and use of the assessment measures. Regular supervision meetings will be arranged for research staff and therapists for case discussions, identifying and managing distress in participants. A study protocol will be in place prior to the start of the trial. This will include details about how to manage difficult situations arising in research, safety and lone working arrangements.

## Treatment as usual

Local psychiatric, medical and primary care services provide standard routine care according to their clinical judgement and available resources. We will administer Client Services Receipt Interview (CSRI)[29] to obtain details regarding treatment received by each study participant. These participants will receive an initial assessment along with TAU as ascertained by their treating doctor at the hospital or their general practitioner (GP). The current practice is that self-harm patients are not routinely referred to mental health services. Research staff will record the nature and intensity of the routine care delivered for each participant.

## PPIE group

This group will help to ensure that the research agenda for self-harm and suicide prevention studies is informed by and aligned with service users and carer priorities.

## Statistical analysis

Statistical analysis will be based on intention-to-treat principles, subject to the availability of data. During the course of the trial, periodic random quality checks of data will be carried out by the trial statistician blind to treatment allocation. Once data entry has been completed, preliminary data analysis will be carried out blind to treatment allocation, prior to unblinding. The results of the trial will follow the standard Consolidated Standards of Reporting Trials recommendations. Baseline and follow-up data will be summarised using the appropriate descriptive statistics and graphical summaries. Treatment effects will be presented with 95% CIs. We will investigate baseline factors that predict nonresponse using a logistic random effects model as non-response may be clustered by therapy group. The statistical analysis of the primary outcome measure (repetition of the self-harm episode within 12 months) will be based on a logistic random effects model with randomised treatment, age, gender and type of self-harm as fixed effects and a random effect of therapist in the treatment arm, with individuals in the control arm being treated as clusters of size 1. The continuous secondary outcome measures will calculate treatment effects using a linear mixed model, with a random effect of therapist and the set of baseline covariates as above including baseline values of the outcome where available. A single model will be fitted across all time points, with a fixed effect for time, and interactions between time and treatment group. There will be no adjustment to secondary outcomes CIs for multiple testing. Binary secondary outcomes will take a similar approach to continuous outcomes, but will use logistic random effects models.

## Economic evaluation

The economic evaluation will take a societal perspective that incorporates the costs of both formal and informal healthcare and other relevant economic impacts such as on education. It will include financial impacts on providers and out of pocket payments by participants and their families. The use and payment for services will be collected using the CSRI[29] with patients and their families at baseline and at each follow-up assessment. The use of the YCMAP intervention will be recorded separately by the trial team. Appropriate unit costs will be collected and attached to individual-level resource use quantities to estimate total care costs for each individual. Cost estimates for the YCMAP intervention will include (1) the salary of the therapists and (2) indirect costs like session preparation, supervision, on-costs and capital overheads. Comparisons of total costs between the YCMAP and TAU arms will be based on non-parametric bootstrapped regressions (with covariates for baseline costs, outcome

measures and other key baseline factors) to account for the likely non-normal distribution in costs. An inital analysis will examine whether any additional costs associated with the YCMAP intervention are offset by savings elsewhere.

More formal assessments of cost-effectiveness will link between-arm differences in average costs with between-arm differences in (1) the primary outcome measure, repetition of self-harm (SASII) and (2) quality-adjusted life-year (QALY) gains. QALYs will be estimated by applying relevant utility weights to health states measured through EuroQol-5 Dimensions-Youth (EQ-5D-Y).[23 30] Costs and outcomes will be linked in the form of incremental cost-effectiveness ratios where relevant, and cost-effectiveness acceptability curves based on the net benefit approach. As per the clinical effectiveness analyses, we will use an intention-to-treat approach for all economic analyses.

## Qualitative data analysis

A purposefully selected subset (stratified by age, gender and self-harm severity) of participants in the treatment group will be invited to complete a qualitative one-to-one digitally recorded interview (face to face or via phonecall) lasting 1–1.5 hours. This will focus on their experience with the YCMAP intervention, the barriers and facilitators to engagement and perceived positive or negative experience with the intervention. A sample of 12–20 participants is likely to be sufficient to ensure data saturation.[31 32] Focus groups (k=2 with 8–10 participants each) with therapists and key stakeholders will also be conducted to enable a further process analysis concerning the wider implementation of YCMAP into the Pakistani health systems. During both the individual and group interviews, broader suggestions for management and prevention of self-harm among young Pakistanis will also be collected to inform future service and research development.

We will explore perspectives of therapists and their supervisors about training and delivery of the model of care. We will also interview other key stakeholders to establish facilitators and barriers to implementation of the intervention in Pakistan. Targeted interviews will be done with a sample of participants who completed the intervention and a sample of participants who 'dropped out' before completion. Up to 30 interviews are likely to be needed with patient participants, to achieve category saturation. New text mining tools will be used to link heterogeneous data, to ensure that knowledge scattered with in lengthly textual data reaches clinicians, patients and policy-makers to support decision making using different sources of information.[33] Qualitative data will be analysed using thematic analysis[34] adopting a critical realist perspective.[35] This will enable the research team to draw conclusions on the intervention while accounting for contextual and cultural factors.

Topic guides will be developed through discussion among trained qualitative researchers as well as with reference to relevant literature. Semistructured interviews will be done to explore views on the 'effectiveness and

'sustainability' of the YCMAP intervention in the management of people presenting with self-harm. Written or verbal consent at the time of interview (face to face or telephone) or prior to the interview (considering current pandemic) will be obtained from study participants and stake holders.

### Internal pilot study

We have included an internal pilot phase (initial 12 months) with clear stop/go criteria across the study centres. This is included to ensure achievement of recruitment targets is viable across all sites included in this full trial. Progression to the full trial will be dependent on meeting the key objectives and stop/go criteria outlined below. The pilot phase will examine the processes for recruitment and logistical practices to support the effective execution of the full RCT. The internal pilot phase will be conducted in line with the National Institute of Health Research guidelines.[36] The key objectives of the pilot are as follows: (1) Recruitment of sufficient number of GPs/Hospital Departments to support the trial and (2) Identify the proportion of eligible participants recruited across the study centres, and examine the feasibility of achieving recruitment targets to a full multicentre RCT. The pilot will be judged successful if sufficient GPs and hospitals are recruited to enrol 200 participants in the first year of recruitment, that is, 4–12 months. We will monitor recruitment and identify the reasons for any shortfall on a monthly basis. The predetermined Stop-Go criteria is as follows:

1. Go if successfully recruit 180 (target minus 10%) or more participants during pilot study phase.
2. Rescue plan to be implemented if less than 180 but greater than 120 participants recruited in pilot phase.
3. Stop if less than 120 participants recruited into the trial.

### Theory of change

The study team uses theory of change (ToC) as the standard framework for all research projects. The overall study will be underpinned by the ToC causal model of planning, monitoring, evaluation and impact assessment to ensure that marginalised voices are included in developing the vision, identifying barriers and challenges through the lens of the end beneficiaries and short-term, medium-term and long-term outcomes are developed and delivered so real change happens from the perspective of the target group. A ToC workshop with key stakeholders has already taken place to ensure all stakeholders have a buy in and ownership of the process (see figure 2).

The ToC causal diagram as an output is evidence of stakeholder engagement and ownership. Also, ToC will enable us to show how trial results are adopted in wider practice and on what basis,what barriers and challenges were faced during the trial and what assumptions were made in defining the goal statements? Informed by the ToC workshop, a Young People's Advisory Group will be set up to provide insight into the lived experiences of today's youth in Pakistan.

As religiosity was a dominant thread from the community engagement exercise, there was recognition that spiritual advice was often the first point of contact for patients rather than health professionals. The trial was not designed to try to replace such religious norms but to work alongside them while recognising that there could be delays in referrals. There was also no restriction on accessing spiritual guidance making professional psychological therapies an addition not a substitute for such support. Importantly, mental health concerns are stigmatising and suicide is still considered as a criminal offence.

Working with religious leaders was therefore a critical part of community engagement with Y-CMAP and its evaluation through the RCT the concept of 'dawa and

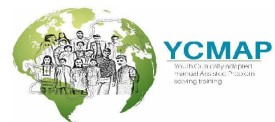

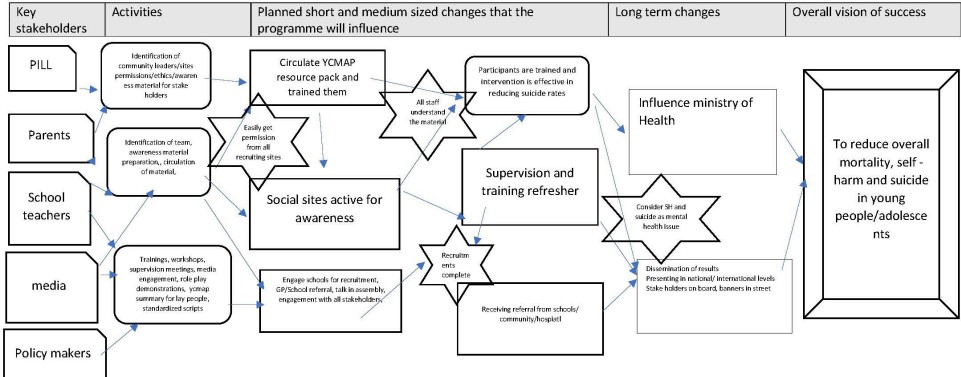

**Figure 2** Theory of change. GP, general practitioner; PILL, Pakistan Institute of Living and Learning;SH, Self-Harm.

dua' medicine and prayer going hand in hand as part of therapy tend to promote adherence and engagement.

## COVID-19 continuity plan

Due to global pandemic COVID-19, amendments to study procedures in response to the lock down and social distancing in Pakistan were made to ensure smooth running of the project (see online supplemental file).

## Trial steering committee

We will convene an independently chaired trial steering committee (TSC) to approve and provide oversight of the trial throughout its various stages. The TSC will include the chair, user representative, three PIs, an independent statistician and a representative from the local health department. This will be independent of the trial management team. The (TSC) will convene annually, but twice in the first year, with feedback from the chair as and when needed.

## Data monitoring and ethics committee

The data monitoring and ethics committee will oversee the data and advise the TSC on any ethical or safety concerns.

## Ethics and dissemination

Ethics approval for the trial has been obtained from the Research Ethics Committee of University of Manchester (Ref: 2019-5024-10755) and the National Bioethics Committee in Pakistan (Ref: No.4-87/NBC-419/19/1213). The study findings will be disseminated using social media, at national and international conferences in partnership with the 'service user', 'carer' and 'community organisations'. Also, study results will be submitted to peer and non peer-reviewed journals for publication.

## DISCUSSION

There has been a recent significant increase in number of trials related to self-harm and suicide prevention which reflects the need and international concern about self-harm/suicide prevention. However, in Pakistan, evidence-based research is limited with no specific intervention for adolescents with self harm history. The evidence suggests that CMAP intervention for adults is effective in Pakistan.[15] The components of CMAP intervention may also be beneficial for adolescents presenting after self-harm, therefore, there is a strong need to determine the effectiveness of culturally adapted manual assisted problem solving intervention for adolescents in Pakistan. Findings will also contribute to evidence based treatments for self-harm and will meet important clinical needs of Pakistan and will help guide policy makers in developing suicide prevention policies.

## Author affiliations

[1]Division of Psychology and Mental Health, The University of Manchester School of Medical Sciences, Manchester, UK

[2]Manchester Global Foundation, Manchester, UK
[3]Division of Neuro-Cognitive Disorder, Older Adults Mental Health, Pakistan Institute of Living and Learning, Karachi, Pakistan
[4]Psychiatry, Greater Manchester West Mental Hlth NHS Fdn Trust, Manchester, UK
[5]Psychiatry, Dr Ziauddin Hospital, Karachi, Sindh, Pakistan
[6]Division of Mood Disorder, Pakistan Institute of Living and Learning, Karachi, Pakistan
[7]Psychology & Mental Health, The University of Manchester, Manchester, UK
[8]Psychiatry, University of Glasgow, Glasgow, UK
[9]Science and Technology Studies, University College London, London, UK
[10]Psychiatry, SRCC Children's Hospital (Narayana Health), Mumbai, India
[11]Health Economics, Pakistan Institute of Living and Learning, Karachi, Pakistan
[12]School of Computer Science, The University of Manchester, Manchester, UK
[13]Psychiatry, Sir Cowasjee Jehangir Institute, Hyderabad, Pakistan
[14]Division of Psychology and Mental Health, The University of Manchester, Manchester, UK
[15]Bradford Teaching Hospitals NHS Foundation Trust, Bradford, UK
[16]Psychiatry, Services Institute of Medical Sciences, Lahore, Punjab, Pakistan
[17]Psychology, Middlesex University, London, UK
[18]Psycho-social Interventions, University of Glasgow, Glasgow, UK
[19]Research Ethics and Governance, University College London, London, UK
[20]Department of Biostatistics and Health Informatics, Institute of Psychiatry, Psychology and Neuroscience, King's College London, London, UK
[21]Division of Neuroscience & Experimental Psychology, The University of Manchester, Manchester, UK
[22]Health Economics, Hallam University, Sheffield, UK
[23]Psychiatry, Centre for Suicide Research, Oxford University, Oxford, UK
[24]Community Medicine, Karachi Medical and Dental College, Karachi, Pakistan
[25]Division of Eating Feeding, Nutrition and Elimination Disorders, Pakistan Institute of Living and Learning, Karachi, Pakistan
[26]Division of Population Health, Health Services Research & Primary Care, Manchester Institute for Collaborative Research on Ageing, University of Manchester, Manchester, UK
[27]Lifestyle & Wellness, UK, UK
[28]Psychiatry, The Tree House, Rawalpindi, Pakistan
[29]Psychiatry, University of Toronto, Toronto, Ontario, Canada
[30]Psychiatry, Dow University of Health Sciences, Karachi, Pakistan
[31]Division of Child and Adolescent Mental Health, Pakistan Institute of Living and Learning, Karachi, Pakistan
[32]Division of Population Health, Health Services Research & Primary Care, University of Manchester, Manchester, UK
[33]Health Economics, Queen Mary University, London, UK
[34]Center for Primary Care and Health Services Research, The University of Manchester, Manchester, UK
[35]Health Economics, Liverpool School of Tropical Medicine, Liverpool, UK
[36]Child and Adolescent Psychiatrist, South London and Maudsley NHS Foundation Trust, London, UK
[37]Department of Applied Psychology, Bahauddin Zakariya University, Multan, Punjab, Pakistan
[38]Psychiatry, Institute of Psychiatry, Rawalpindi, Pakistan
[39]Child and Adolescent Mental Health, Pakistan Institute of Living and Learning, Karachi, Pakistan
[40]Nursing, Iqra University, Karachi, Pakistan
[41]Research and Development, Pakistan Institute of Living and Learning, Karachi, Pakistan

**Acknowledgements** We express our gratitude to all team members, researchers, consultants and investigators for their support and advice in the development and setting up of the trial. We would like to thank community gate keepers, service uers and patient and public involvement group members for their kind support, input and feedback on the development of the funding application, spreading awareness among masses and in trial recruitments.We would also like to thank Trial Steering Committee (TSC) members, Prof. David Kingdon (Chair), Dr. Mehmood Shaukat, Faraz Usman and Data Management and Ethics Committee (DMEC) members Professor Unaiza Niaz (Chair), Dr. Qamar Saeed and Dr. Maryum Ilyas for overseeing the trial's progress and management.

**Contributors** NH, NC, TK, IBC and ST involved in conceptualisation, designing and planning of the project (including protocol development). EC and SAg reviewed

qualitative component. ZZ, CW, PT, KD and FN reviewed intervention section, RM facilitated the ToC process. MAA and RE reviewed statistical analysis plan and APo and AG reviewed economic evaluation plan. All authors SAs, SAu, MHA, AB, SAI, SE, JG, KH, FJ, AK, TM, AMc, AMi, HAN, AN, MP, APa, TS, MS, SS, ATN and SNZ gave their valuable inputs in finalising the study protocol.

**Funding**  The Medical research Council /DFID/NIHR programme (MR/R022461/1) has funded this trial.

**Competing interests**  NH, former Trustee of 'Pakistan Institute of Living and Learning (PILL)', 'Abaseen Foundation (UK)' and 'Lancashire Mind (UK)'. At 'Manchester Global Foundation', he is the Chair of Board of Trustees. He is also a member of the executive committee for the Faculty of Academic Psychiatry, at the Royal College of Psychiatrists. NH is a NIHR Senior Investigator. NC is Associate Director of Global Mental Health and Cultural Psychiatry Research Group. IBC, former Trustee of 'PILL' is Honorary Professor at the University of Manchester. NH, IBC and NC have received support for educational programmes and/or travel support and/or speaker fees from pharmaceutical companies. CW is the author of a book aimed at suicide prevention, and has written a range of books and online CBT-based course resources that are available as both, free access and on a commercial basis. NH, NC, IBC and TK's time is partially funded by the Global Challenges Research Fund 'South Asia Harm Reduction Movement-SAHAR M' (MR/P028144/1).

**Patient and public involvement**  Patients and/or the public were involved in the design, or conduct, or reporting, or dissemination plans of this research. Refer to the Methods section for further details.

**Patient consent for publication**  Not applicable.

**Provenance and peer review**  Not commissioned; externally peer reviewed.

**ORCID iDs**
Nusrat Husain http://orcid.org/0000-0002-9493-0721
Tayyeba Kiran http://orcid.org/0000-0003-2478-4148
Christopher Williams http://orcid.org/0000-0001-5387-2863
Erminia Colucci http://orcid.org/0000-0001-9714-477X
Keith Hawton http://orcid.org/0000-0003-4985-5715
Thomas Mason http://orcid.org/0000-0003-3135-0364

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
