## [Reviewer comments · BMJ Open]

ARTICLE DETAILS

TITLE (PROVISIONAL)	A Youth Culturally adapted Manual Assisted Problem Solving Training (YCMAP) in Pakistani adolescent with a history of self harm: Protocol for multi-centre clinical and cost effectiveness randomised controlled trial.
AUTHORS	Husain, Nusrat; Tofique, Sehrish; Chaudhry, Imran B.; Kiran, Tayyeba; Taylor, Peter; Williams, Christopher; Memon, Rakhshi; Aggarwal, Shilpa; Alvi, Mohsin; Ananiadou, S; Ansari, Moin; Aseem, Saadia; Beck, Andrew; Alam, Sumira; Colucci, Erminia; Davidson, Kate; Edwards, Sarah; Emsley, Richard; Green, Jonathan; Gumber, Anil; Hawton, Keith; Jafri, Farhat; Khaliq, Ayesha; Mason, Thomas; McCreath, Ann; Minhas, Ayesha; Naeem, Farooq; Naqvi, Haider; Noureen, Amna; Panagioti, Maria; Patel, Anita; Poppleton, Aaron; Shiri, Tinevimbo; Simic, Mima; Sultan, Sarwat; Nizami, Asad; Zadeh, Zainab; Zafar, Shehla; Chaudhry, Nasim

VERSION 1 – REVIEW

REVIEWER	Dennis Ougrin King's College London, Institute of Psychiatry
REVIEW RETURNED	03-Sep-2021

GENERAL COMMENTS	The authors are planning to mount a very important study. I have the following remarks  1. Could the authors please clarify if they are doing an efficacy or an effectiveness study. If the latter, they need to justify and explain how they are going to do it. 2. There is a discrepancy between the objectives of this study (reduce self-harm repetition) and the objectives stated in the ToC (reduce mortality as well as self-harm) - this needs to be addressed. 3. The authors might like to address the issue of a relatively weak evidence base of CBT and related interventions (other than DBT) in young people, see Kothgassner, O. et al (2020). "Does treatment method matter? A meta-analysis of the past 20 years of research on therapeutic interventions for self-harm and suicidal ideation in adolescents." Borderline Personality Disorder and Emotion Dysregulation 7(1).
---

REVIEWER	Sergey A Igumnov V.Serbosky National Research Centre for Psychiatry and Narcology, Addictology
REVIEW RETURNED	27-Sep-2021

GENERAL COMMENTS	The article "A Youth Culturally adapted Manual Assisted Problem Solving Training (YCMAP) in Pakistani adolescent with a history of self harm: Protocol for multi-centre clinical and cost effectiveness randomised controlled trial".
---

	Priority directions for the implementation of the strategy for self-harm and suicide preventing in Pakistan is of great theoretical and practical importance, since the problems associated with a high level of self-harm and suicide attempts by the adolescent population pose a real threat to demographic security and social stability of any state. The problems associated with a high level of self-harm and suicide attempts by the population remain relevant for many countries of the world, including the Republic of Belarus. In the Republic of Belarus, work on the prevention and overcoming of self-harm and suicide attempts by the adolescent population at the state level is carried out in the course of the implementation of measures of the state programs "People's health and demographic security of the Republic of Belarus". One of the important strategies for the prevention of self-harm and suicide attempts by the adolescent population are measures that motivate the population to maintain a healthy lifestyle, which helps to reduce self-harm and suicide attempts by the adolescent population and, as a result, reduce the negative consequences associated with it. This work is of undoubted practical interest for healthcare organizers and people involved in preventive medicine, is recommended for publication and can be published in the open press. Doctor of Medical Sciences, Professor of the Institute of Psychology of Belarusian State Pedagogical University S. Igumnov
--	--

REVIEWER	Alberto Forte Sapienza University of Rome
REVIEW RETURNED	27-Sep-2021

GENERAL COMMENTS	I found the present study protocol is very well written and complete. The topic is timely and of interest, as suicidal behavior data in developing countries is under-reported. The background is presented clearly. My main concern regards the use of the term self-harm. It is not clear to me whether the project includes only suicide attempts and/or non-suicidal self-injuries (NSSI). I understand it could be a matter of terminology, but it would be important to clarify the outcome and also justify more clearly whether the intervention is aimed to treat only suicide attempts or if it includes NSSI. I have also a few minor concerns:  - Exclusion criteria are vague, what do you mean you exclude patients with severe psychiatric conditions? are only psychotic disorders excluded? how about Bipolar disorders? - I am not sure it is good for readers to present methodological aspects in bullets, also check if it is appropriate for the preparation of the MS in this journal
--

REVIEWER	Alex Millner Harvard University, Department of Psychology
REVIEW RETURNED	01-Oct-2021

GENERAL COMMENTS	This paper describes a proposed randomized controlled trial to test a Manual Assisted Problem Solving Training to reduce self harm among Pakistani youth. I do not have any of the comments or suggestions. The paper is organized, clear, well-written and thoroughly explains in detail the proposed procedures and rationale for the study. I wish the authors luck.
--

VERSION 1 – AUTHOR RESPONSE

Reviewer: 1

Dr. Dennis Ougrin, King's College London

Comments to the Author:

The authors are planning to mount a very important study. I have the following remarks

1. Could the authors please clarify if they are doing an efficacy or an effectiveness study. If the latter, they need to justify and explain how they are going to do it.

Response: Primary aim of our research is to determine the clinical effectiveness over 12 months of the culturally adapted Manual Assisted brief psychological intervention (Y-CMAP) which will be measured through our primary outcome measure i.e. repetition of self-harm 12-month post-randomization. The study will also assess the cost effectiveness of Y-CMAP, explore the mechanism through which it works, and to capture participants subjective experience of using the intervention, including what is helpful, unhelpful and any problems or barriers to using the therapy. We will run Intention-to-treat analysis to assess effectiveness. We have now made sure to be consistent in using terms and we have replaced the term “efficacy” in last paragraph of back ground section to “clinical effectiveness”.

2. There is a discrepancy between the objectives of this study (reduce self-harm repetition) and the objectives stated in the ToC (reduce mortality as well as self-harm) - this needs to be addressed.

Response: ToC is an ongoing process which is based on causal model of planning, monitoring, evaluation and impact assessment. This will enable us to show how trial results are adopted in wider practice and on what basis. “Reduction in mortality, self-harm and suicide in young adolescents” is our vision statement that we want to achieve as long term impact. As we know, a vision statement is usually concise so that it can be easily remembered; clear; and inspiring. It should also be challenging—that is, the change we are trying to achieve in the world should be significant enough that it will not be easy to achieve in the short term. So, considering this our stakeholders stated reduction in mortality and self-harm, and suicide as our vision statement which is global target and included as an indicator in the UN Sustainable Development Goals. Also, the vision is aligned with well-established evidence regarding association between reduction in repetition rates with reduction in suicide as self-harm is the single most important predictor of suicide (Griffin et al 2020, Cripps et al 2020).

We have added these lines in ToC

“As religiosity was a dominant thread from the community engagement exercise, there was recognition that spiritual advice was often the first point of contact for patients rather than health professionals. The trial was not designed to try to replace such religious norms but to work alongside them while recognising that there could be delays in referrals. There was also no restriction on accessing spiritual guidance making professional psychological therapies an addition not a substitute for such support. Importantly, mental health concerns are stigmatising and suicide is still considered as a criminal offense”.

Working with religious leaders was therefore a critical part of community engagement with Y-CMAP and its evaluation through the RCT the concept of ‘dawa and dua’ medicine and prayer going hand in hand as part of therapy tend to promote adherence and engagement”.

*3. The authors might like to address the issue of a relatively weak evidence base of CBT and related interventions (other than DBT) in young people, see Kothgassner, O. et al (2020). "Does treatment method matter? A meta-analysis of the past 20 years of research on therapeutic interventions for self-harm and suicidal ideation in adolescents." *Borderline Personality Disorder and Emotion Dysregulation* 7(1).*

Response: Thank you for sharing a recent reference. We have now added following lines into the background section to support our rationale of using CBT

“Considering the benefits of adult individual Cognitive Behaviour Therapy (CBT)-based psychotherapy (KG Witt et al, 2020), tested in our previous trial in Pakistan with positive results (Husain et al 2014) and replicated in our recently completed trial with 901 adults participants in which we had to exclude 265 adolescents (less than 18 years) (CMAP2) (Kiran T et al 2021). An unmet need was recognized to develop CMAP further for adolescent and adapting the similar brief psychological intervention for children and adolescents.

Reviewer: 2

Dr. Sergey A Igumnov, V.Serbsky National Research Centre for Psychiatry and Narcology, Belarusian State Pedagogical University

Comments to the Author:

The article “A Youth Culturally adapted Manual Assisted Problem Solving Training (YCMAP) in Pakistani adolescent with a history of self-harm: Protocol for multi-centre clinical and cost effectiveness randomised controlled trial”.

Priority directions for the implementation of the strategy for self-harm and suicide preventing in Pakistan is of great theoretical and practical importance, since the problems associated with a high level of self-harm and suicide attempts by the adolescent population pose a real threat to demographic security and social stability of any state.

The problems associated with a high level of self-harm and suicide attempts by the population remain relevant for many countries of the world, including the Republic of Belarus.

In the Republic of Belarus, work on the prevention and overcoming of self-harm and suicide attempts by the adolescent population at the state level is carried out in the course of the implementation of measures of the state programs “People's health and demographic security of the Republic of Belarus”.

One of the important strategies for the prevention of self-harm and suicide attempts by the adolescent population are measures that motivate the population to maintain a healthy lifestyle, which helps to reduce self-harm and suicide attempts by the adolescent population and, as a result, reduce the negative consequences associated with it.

This work is of undoubted practical interest for healthcare organizers and people involved in preventive medicine, is recommended for publication and can be published in the open press.

Doctor of Medical Sciences, Professor of the Institute of Psychology of Belarusian State Pedagogical University S. Igumnov

Response: We are thankful to the reviewer, Dr. Sergey A Igumnov, for very encouraging comments.

Reviewer: 3

Dr. Alberto Forte, Sapienza University of Rome

Comments to the Author:

I found the present study protocol is very well written and complete. The topic is timely and of interest, as suicidal behavior data in developing countries is under-reported. The background is presented clearly.

My main concern regards the use of the term self-harm. It is not clear to me whether the project includes only suicide attempts and/or non-suicidal self-injuries (NSSI). I understand it could be a

matter of terminology, but it would be important to clarify the outcome and also justify more clearly whether the intervention is aimed to treat only suicide attempts or if it includes NSSI.

Response: We defined self-harm in our inclusion criteria as: *“an act with non-fatal outcome, in which an individual deliberately initiates a non-habitual behaviour that, without interventions from others, will cause self-harm, or deliberately ingests a substance in excess of the prescribed or generally recognised therapeutic dosage, and which is aimed at realizing changes which the subject desired via the actual or expected physical consequences”*

Thus, people with non-suicidal self-injuries (NSSI) do not fulfill the inclusion criteria for this trial.

I have also a few minor concerns:

- Exclusion criteria are vague, what do you mean you exclude patients with severe psychiatric conditions? are only psychotic disorders excluded? how about Bipolar disorders?*
- I am not sure it is good for readers to present methodological aspects in bullets, also check if it is appropriate for the preparation of the MS in this journal*

Response: We have now including the following in the exclusion criteria to make it clearer. “Patients with a severe mental illness such as **schizophrenia spectrum and other psychotic disorders**, bipolar and severe depressive disorder will be excluded”. We consider that patients with severe psychiatric conditions require a different approach and we are considering this for a future study in Pakistan.

Reviewer: 4

Dr. Alex Millner, Harvard University

Comments to the Author:

This paper describes a proposed randomized controlled trial to test a Manual Assisted Problem Solving Training to reduce self-harm among Pakistani youth. I do not have any of the comments or suggestions. The paper is organized, clear, well-written and thoroughly explains in detail the proposed procedures and rationale for the study. I wish the authors luck.

Response: We are thankful to Dr. Alex Millner for very encouraging comments.

VERSION 2 – REVIEW

REVIEWER	Dennis Ougrin King's College London, Institute of Psychiatry
REVIEW RETURNED	08-Dec-2021

GENERAL COMMENTS	The authors addressed my remarks well. I have no further remarks.
---

REVIEWER	Alberto Forte Sapienza University of Rome
REVIEW RETURNED	26-Dec-2021

GENERAL COMMENTS	Thanks for asking me to review the present study protocol. I found it very interesting and the topic is timely and of interest. There's an urgent need for new strategies and therapies to treat adolescents who self-harm, particularly in several countries. I think the protocol is well written and well organised, the methodology is clearly described. I would recommend minor changes as a suggestion to improve the quality of the protocol:
---

	- I do not understand why patients with severe mental illness and severe mood disorders are excluded; those are at the highest risk and would benefit from new specific treatments together with pharmacological therapies. I would suggest specifying this aspect and in case justify better the exclusion, as well as specifying better to whom the treatment is suggested. - I would talk more about the importance of addressing self-harm in adolescents. There are provisional data on the increase in self-harm among girls in the USA, and we are living the same exponential increase in Europe, I would mention this in the introduction to highlight the importance of this.
--	---

VERSION 2 – AUTHOR RESPONSE

Reviewer: 1

Dr. Dennis Ougrin, King's College London

Comments to the Author:

The authors addressed my remarks well. I have no further remarks.

Reviewer: 3

Dr. Alberto Forte, Sapienza University of Rome

Comments to the Author:

Thanks for asking me to review the present study protocol. I found it very interesting and the topic is timely and of interest. There's an urgent need for new strategies and therapies to treat adolescents who self-harm, particularly in several countries. I think the protocol is well written and well organised, the methodology is clearly described. I would recommend minor changes as a suggestion to improve the quality of the protocol:

- I do not understand why patients with severe mental illness and severe mood disorders are excluded; those are at the highest risk and would benefit from new specific treatments together with pharmacological therapies. I would suggest specifying this aspect and in case justify better the exclusion, as well as specifying better to whom the treatment is suggested

Response: We have now added following lines in our 'exclusion criteria' to support our rationale of excluding people with severe mental illnesses.

"Patients with a severe mental illness such as **schizophrenia spectrum and other psychotic disorders**, bipolar and severe depressive disorder will be excluded". This intervention has been developed to help individuals who experience common mental health problems such as anxiety or mild to moderate depressive disorders. It has not been developed to work with people with severe mental health difficulties such as psychotic disorders. We consider that the intervention would require work with service users and carers and further adaptation work with people who have severe mental health illness.

- I would talk more about the importance of addressing self-harm in adolescents. There are provisional data on the increase in self-harm among girls in the USA, and we are living the same exponential increase in Europe, I would mention this in the introduction to highlight the importance of this.

Response: We have added following lines in our 'Introduction' section.